# Severe Dyslipidemia Mimicking Familial Hypercholesterolemia Induced by High-Fat, Low-Carbohydrate Diets: A Critical Review

**DOI:** 10.3390/nu15040962

**Published:** 2023-02-15

**Authors:** Veera Houttu, Aldo Grefhorst, Danny M. Cohn, Johannes H. M. Levels, Jeanine Roeters van Lennep, Erik S. G. Stroes, Albert K. Groen, Tycho R. Tromp

**Affiliations:** 1Department of Vascular Medicine, Amsterdam UMC, Location AMC, Meibergdreef 9, 1105 AZ Amsterdam, The Netherlands; 2Department of Experimental Vascular Medicine, Amsterdam UMC, Location AMC, Meibergdreef 9, 1105 AZ Amsterdam, The Netherlands; 3Department of Internal Medicine, Erasmus MC, University Medical Center, Doctor Molewaterplein 40, 3015 GD Rotterdam, The Netherlands

**Keywords:** carnivore diet, ketogenic diet, low-carbohydrate diet, high-fat diet, LDL cholesterol, familial hypercholesterolemia, cardiovascular disease

## Abstract

Emerging studies in the literature describe an association between high-fat, low-carbohydrate diets and severe hypercholesterolemia consistent with the levels observed in patients with (homozygous) familial hypercholesterolemia (FH). High levels of low-density lipoprotein cholesterol (LDL-C) may result from the reduced clearance of LDL particles from the circulation, the increased production of their precursor, or a combination of both. The increased intake of (saturated) fat and cholesterol, combined with limited to no intake of carbohydrates and fiber, are the main features of diets linked to hypercholesterolemia. However, several observations in previous studies, together with our observations from our lipid clinic, do not provide a definitive pathophysiological explanation for severe hypercholesterolemia. Therefore, we review these findings and possible pathophysiological explanations as well as opportunities for future research. Altogether, clinicians should rule out high-fat, low-carbohydrate diets as a possible cause for hypercholesterolemia in patients presenting with clinical FH in whom no mutation is found and discuss dietary modifications to durably reduce LDL-C levels and cardiovascular disease risk.

## 1. Introduction

The notion that circulating low-density lipoprotein cholesterol (LDL-C) causes atherosclerotic cardiovascular disease (ASCVD) is firmly rooted in evidence from genetic, epidemiological, and clinical studies [1]. Interestingly, some of the earliest lines of evidence were derived from experimental studies involving the modification of plasma cholesterol levels through diet. The pioneering groundwork for cholesterol’s involvement in atherosclerosis can be traced back to St. Petersburg in 1913, when a young experimental pathologist named Anitschkow induced atherosclerosis in rabbits by feeding them purified cholesterol dissolved in sunflower oil. Control animals fed only the sunflower oil showed no lesions [2]. In the following decades, evidence accumulated that unraveled the relationship between (dietary) cholesterol, atherosclerosis, and potential options for treatment [3]. Over a century after Anitchkow’s seminal observations, the interplay between diet and (severe) dyslipidemia is still of special scientific interest, specifically concerning high-fat, low-carbohydrate diets, which is the topic of the narrative review presented here.

Epidemiological observations in large, cross-cultural prospective studies from the general population have been instrumental in linking the intake of fat and cholesterol with plasma cholesterol levels and the incidence of coronary events [4,5]. The first recommendations on dietary intervention to prevent cardiovascular disease were proposed by large professional medical associations in the 1960s [6], and it has now been widely acknowledged that diet has a modest yet meaningful influence on LDL-C levels on a population level [7,8]. The first data to show that cholesterol uptake in the intestine was amenable to a pharmacotherapeutic intervention were derived from the Coronary Primary Prevention Trial, published in 1984. This landmark placebo-controlled randomized clinical trial showed that the bile acid sequestrant cholestyramine, which helps remove cholesterol from the enterohepatic circulation, lowered LDL-C levels and reduced cardiovascular events in 3806 asymptomatic middle-aged men with primary hypercholesterolemia [9].

Bile acid sequestrants have since then been surpassed by statins and other more effective pharmacological lipid-lowering therapies, but dietary modifications continue to be highlighted in the most recent clinical guidelines for the prevention of cardiovascular disease [10,11,12], partly due to their LDL-lowering effect [7,13]. More specifically, guidelines recommend plant-based and Mediterranean-type diets containing vegetables, nuts, whole grains, and fish rich in unsaturated fats and dietary fibers, while the intake of processed meat, deep-fried foods, ice cream, high-fat dairy products, refined carbohydrates, and sweetened beverages high in saturated fatty acids (SFAs), cholesterol, trans-fatty acids (TFAs), sodium, and glucose should be avoided or kept to a minimum.

Besides observational and interventional data on cholesterol intake or reuptake in large study populations, understanding the link between plasma LDL-C, atherosclerosis, and treatment thereof has hugely progressed through the study of patients with extremely elevated plasma LDL-C levels due to genetic causes [3,14]. Patients with familial hypercholesterolemia (FH), an autosomal dominant inherited disorder caused by variants in the genes involved in lipoprotein metabolism [15], have elevated LDL-C levels from birth, which are more than double the LDL-C levels observed in patients from the general population [16,17]. In general, such high LDL-C levels are not observed in healthy individuals who adhere to a dietary pattern rich in fat and cholesterol.

Sparked by striking observations from our own clinical practice, we review the available evidence of high-fat, low-carbohydrate diets linked to severe dyslipidemia that may be mistaken for (clinical) FH. We discuss this phenomenon in light of possible pathophysiological explanations, point out knowledge gaps for future research, and provide practical recommendations to clinicians who encounter such patients.

## 2. High-Fat, Low-Carbohydrate Diets: Examples from the Lipid Clinic

### 2.1. Carnivorous Diet

Two brothers aged 33 years old (patient 1) and 28 years old (patient 2) were referred by their general practitioner to a university lipid clinic because of severe hypercholesterolemia. Upon referral, the LDL-C levels reported by the general practitioner were 15 and 12 mmol/L, respectively.

Both patients had an unremarkable medical history, used no medication, and had no family history of dyslipidemia or (premature) cardiovascular disease. Patients reported exercising regularly (approximately 4×/week, including resistance training) and had a muscular physique with a body mass index (BMI) of 26.2 and 27.3, respectively.

Physical examination was normal and did not specifically show visible cholesterol depositions in the form of xanthomas, corneal arcus, or xanthelasmata.

A fasting blood sample was obtained, and laboratory evaluations showed elevated LDL-C of 15.02 mmol/L and 8.59 mmol/L (patients 1 and 2, respectively, Table 1). We ruled out hypothyroidism and proteinuria as secondary causes of hypercholesterolemia. Liver enzymes were within the upper limits of normal. We suspected familial hypercholesterolemia (FH), possibly even in a homozygous form, as the underlying genetic cause and performed the next-generation sequencing of 27 lipid-related genes [16]. However, no pathogenic variants in any of these genes were found.

Dietary anamnesis revealed that both patients had started a carnivorous diet approximately one year prior to presentation to the lipid clinic. This diet consists solely of meat (mostly red) and high-fat dairy products. The calculated energy intake from carbohydrates on a typical day was <3 E-% (6 g/d), meaning that this diet was strongly ketogenic. The intake from protein and fat was 37 E-% and 61 E-%, respectively (see Appendix A).

Although no lipid profiles were available prior to the initiation of the carnivore diet, we suspected this diet to cause an abnormal lipid profile. We strongly advised the patients to adopt a regular, balanced dietary pattern (including carbohydrates and vegetables) and explained the risks of prolonged exposure to such elevated LDL-C levels. However, both decided to continue the carnivore diet and decided against starting lipid-lowering therapy as an alternative way to reduce the elevated LDL-C levels.

We performed fast protein liquid chromatography (FPLC) using stored plasma. This showed disproportionately elevated very-low-density lipoprotein (VLDL) and intermediate-density lipoprotein (IDL) fractions, which suggested that the overproduction of VLDL contributed to hypercholesterolemia (Figure 1).

To better understand the effect of this diet on glucose metabolism, we measured fasting levels of glucose, insulin, and c-peptide and calculated the HOMA-IR to gauge the level of insulin sensitivity. Insulin and c-peptide were close to the lower limit of normal, and HOMA-IR was found to approximate >95th percentile, compared with a relevant reference population [18], which suggested good to very good insulin sensitivity.

We investigated whether a copious intake of dietary fatty acids and cholesterol would overwhelm the hepatic triglyceride and cholesterol pools and manifest as hepatic steatosis. We performed vibration-controlled transient elastography (VCTE) using FibroScan^®^ 530 Compact (Echosens, France) estimated liver stiffness measurement (LSM) and controlled attenuation parameter (CAP) in the right liver lobe of both patients but found no signs of liver fibrosis or steatosis. Liver MR spectroscopy subsequently confirmed the near absence of liver fat in both patients.

We performed a carotid ultrasound, which revealed intima–media thickening suggestive of the early development of atherosclerosis.
nutrients-15-00962-t001_Table 1Table 1Summary of laboratory and imaging results derived from patients 1 and 2 adhering to a carnivore diet.
Patient 1Patient 2Reference RangeBMI (kg/m^2^)26.227.318.5–25Total cholesterol (mmol/L)16.8110.88
HDL cholesterol (mmol/L)1.532.00
LDL cholesterol (mmol/L)15.028.59
Triglycerides (mmol/L)0.570.65<1.80ApoB (mg/dL)321186<120ApoA1 (mg/dL)156188
Lp(a) (nmol/L)187<105Glucose (mmol/L)4.35.0<6.0Insulin (pmol/L)16.417.512–96C-peptide (nmol/L)0.190.190.2–0.8HOMA-IR0.50.6<2.0FibroScan CAP (dB/m)185185156–288FibroScan LSM (kPa)6.26.2<7Liver fat content (%)<22.7<5Carotid IMT thickness (percentile)>97th90thBased on age and sex, according to [19]


### 2.2. Journey from Zero-Carb to Raw-Food Diet

Patient 3 was a 23-year-old male referred to the academic hospital lipid clinic for the analysis of severe hypercholesterolemia, with an LDL-C level of 12.2 mmol/L reported by his general practitioner.

The patient had no relevant medical history and used no medication. Besides his maternal grandmother, who had started statin therapy at an elderly age, it was unknown if family members had hypercholesterolemia. The family history of cardiovascular disease was negative. The patient told us he had specially requested a lipid panel measurement from his general practitioner because he was interested to learn his blood cholesterol levels after having recently adopted a new diet. The patient reported that, since his youth, he had struggled to maintain a healthy and balanced diet and that he had experienced that his dietary pattern was closely linked to his psychological well-being. The patient had, therefore, experimented with progressively eliminating food products from his diet. This led to the point that, at the time of referral, the patient was on a zero-carb diet rich in animal proteins and fats in the form of eggs and bacon, supplemented with raw minced meat, raw liver, and a limited amount of vegetables.

Physical examination showed a lean young male (BMI 21 kg/m^2^) without signs of xanthoma, xanthelasmata, or corneal arcus, and was otherwise unremarkable.

A repeat lipid profile was obtained, which showed somewhat lower, albeit still severely elevated, levels of LDL-C at 7.9 mmol/L, which could well be consistent with heterozygous FH. However, genetic analysis using next-generation sequencing showed no FH-causing variant. It was suspected that the patient’s diet contributed to the elevated LDL-C levels. Dietary changes as well as the lowering of LDL-C by starting statin therapy were discussed, and the patient opted to modify his diet first.

Three months later, the patient had significantly reduced his intake of animal fats (mostly cut his intake of eggs and bacon), and a repeat lipid panel showed a near normalization of LDL-C levels (Table 2), after which he was referred back to the care of the general practitioner. The most recent lipid panel was completely normal for age and gender. At this time, the patient reported being on a balanced raw food diet with approximately 400 g of raw meat or fish daily, supplemented with leafy vegetables, nuts, and fruit (usually 2 or 3 mangos daily). The patient reported no intake of starchy carbohydrates.

### 2.3. Heterozygous Familial Hypercholesterolemia Derailed

Patient 4 was a male diagnosed with FH (variant p.R3500Q in *APOB*) at the age of 31 years old, with no relevant medical history. Following the diagnosis of FH, a statin combined with ezetimibe had been prescribed as lipid-lowering therapy, and the patient was regularly followed up by his internist at the lipid clinic.

At the age of 41 years old, the patient started a low-carb high-fat diet along with his spouse, who intended to lose weight. At the latest lipid panel obtained before the patient had started this diet, LDL-C levels were relatively well controlled at 2.9 mmol/L, with the patient taking rosuvastatin 40 mg and ezetimibe 10 mg daily. However, when the patient presented for his next yearly follow-up visit after having adopted his new diet, LDL-C levels were severely elevated at 8.39 mmol/L (Table 3). Adherence to lipid-lowering medication was unchanged, and secondary causes of hypercholesterolemia (hypothyroidism and proteinuria) were ruled out. The estimated untreated LDL-C levels would well be within the range observed in patients with homozygous FH.

The patient was referred to a dietician for dietary assessment and counseling. His diet was shown to be low in carbohydrates and high in dairy fats (Appendix B), and recommendations were made for a more balanced diet. His BMI dropped from 23.4 to 21.4 kg/m^2^ with diet. With the help of his dietician, the patient modified his diet by, amongst others, reducing his intake of dairy fat. At the next follow-up visit, LDL-C had completely normalized to pre-diet levels (Table 3), with unchanged adherence to lipid-lowering therapy.

## 3. Mechanism of Dietary Induced Hypercholesterolemia

Recent observations from our clinical practice add to the emerging literature describing the association between high-fat, low-carbohydrate diets and severe hypercholesterolemia. We reviewed these findings and possible pathophysiological explanations for this phenomenon as well as opportunities for future research. We argue that clinicians should rule out adherence to a high-fat, low-carbohydrate diet in patients presenting with clinical FH.

Several other studies have reported patients consuming a high-fat ketogenic diet who present with high LDL-C levels consistent with FH, which was shown to be reversible with the normalization of the diet. Goldberg and colleagues reported five cases of LDL-C levels ranging from 6.3 to as high as 17.7 mmol/L. Genetic analysis revealed no pathogenic FH variants in any of the patients, although the patient with the highest cholesterol levels was found to have dysbetalipoproteinemia (confirmed by identification of *APOE* E2/E2 genotype) [20]. Schaffer et al. reported three patients with severe hypercholesterolemia (LDL-C levels of 8.9, 11.5, and 15.5 mmol/L), whose LDL-C levels decreased after making dietary modifications [21]. Norwitz and colleagues reported a series of five patients on a ketogenic diet whose LDL-C levels ranged from 6.2 to up to 17.2 mmol/L. All tested negative for an FH-causing variant and after the reported moderate reintroduction of carbohydrates (50–100 g/day), hypercholesterolemia attenuated and even normalized in one patient [22]. The same authors also recently described a 26-year-old male with ulcerative colitis who, in an attempt to relieve its symptoms, initiated a ketogenic diet on which LDL-C levels peaked at 14.1 mmol/L [23]. Interestingly, a coronary CT angiography did not show (non)calcified plaque after two years of exposure to LDL-C levels in a similar range, as seen in patients with homozygous FH [23,24].

The fact that this phenomenon is more widespread than the subject of academic curiosity, only described in rare case descriptions, is exemplified by a recent online survey that collected patient-reported data, including lipid panels, from several hundred adults following a carbohydrate-restricted diet [22]. Of all the 903 participants who participated in the survey, 42% and 22% reported having LDL-C levels higher than 6.5 mmol/L and 8.5 mmol/L, respectively. These values are thresholds in the Dutch Lipid Clinic Network (DLCN) criteria used in the (clinical) diagnosis of FH. Notably, 5% of the respondents entered LDLC levels > 13 mmol/L, which is considered to be consistent with homozygous FH, the most severe form of inherited dyslipidemia, which can cause cardiovascular mortality as early as in childhood if left untreated [24,25].

These observations, combined with the cases we described, are examples that both individuals adhering to these diets as well as their clinicians, should be cautious of their potential to cause or exacerbate severe hypercholesterolemia. High-fat, low-carbohydrate (‘keto’) diets may be considered in a line of fad diets known for exaggerated health claims and are frequently propagated through social media, where health misinformation is widespread, and the quality is difficult to assess [26]. Support for these diets is often provided in the form of opinionated, absolute statements that lack the backing of good-quality evidence highlighting harms and benefits [27,28]. Nevertheless, the number of individuals who follow a high-fat, low-carbohydrate diet is considerable and increasing. This diet is reported among the most frequently followed dietary patterns in the United States, with comparable prevalence to the commonly recommended Mediterranean and dietary approaches to stopping hypertension (DASH) diet [29,30]. There is currently no evidence to support that prolonged exposure to high levels of LDL-C in the context of high-fat, low-carbohydrate diets is not atherogenic (and therefore ‘safe’), which should be balanced against the totality of evidence on atherogenic lipoproteins and their causal role in the development of ASCVD [1].

The observations of extreme hypercholesterolemia described here and by others beg a discussion of possible pathophysiological explanations for this phenomenon. High levels of LDL-C may result from the reduced clearance of LDL particles from the circulation, the increased production of their VLDL precursor, or a combination of both. The increased intake of dietary fat and cholesterol, combined with limited to no intake of carbohydrates, are salient features of the diets consumed by the patients we described. In the following sections, we discuss these factors and their potential roles in the pathophysiology of hypercholesterolemia individually, but it is likely that these factors act in concert.

### 3.1. Increased Intake of Dietary Cholesterol

Cholesterol stores in the human body are in a constant state of flux. Cholesterol in LDL particles either originates from intestinal absorption or de novo synthesis in the liver. After an LDL particle is taken up by the liver, its cholesterol content enters the hepatic cholesterol pool and may either be secreted back into the bloodstream packed in lipoproteins or be excreted in bile directly or after conversion to bile acid. Combined with the intake from diet, this cholesterol is partly taken up by the intestine and transported back to the liver and partly exits the body through feces [31]. It could be hypothesized that the hypercholesterolemia observed in our patients is partly due to their cholesterol-rich diet.

However, cholesterol absorption, synthesis, and biliary excretion appear to be balanced in a way that the dietary intake of cholesterol, under normal circumstances, only modestly translates into the cholesterol levels found in the circulation. For example, vegans consuming 90% less dietary cholesterol than omnivores were found to have plasma LDL-C levels that were only 13% lower [32]. A meta-analysis of intervention studies that supplemented cholesterol through diet showed only moderate increases in LDL-C levels (up to 0.22 mmol/L) [33]. The apparent ‘resilience’ of the plasma cholesterol to high dietary cholesterol intake is exemplified by a case report of an 88-year-old male who ate 25 eggs per day but had a normal plasma LDL-C level of 3.68 mmol/L and no clinically important atherosclerosis. Isotope studies revealed that, compared with 10 healthy controls, the subject had a markedly reduced rate of cholesterol absorption and hepatic cholesterol synthesis but greatly increased synthesis of bile acids [34].

In healthy adults, the average fractional absorption rate of cholesterol in the intestine is approximately 50% but with great inter-individual variation ranging from 20% to 80% [35,36]. Although stable isotope studies were beyond the scope of our study, it is possible that the increased dietary intake of cholesterol exceeded the relative capacity to downregulate its intestinal absorption in our patients. The overload of dietary cholesterol delivered to hepatocytes may contribute to the downregulation of the LDL receptor (LDLR) and thus increase the circulating LDL-C levels [37,38,39].

The influence of gut microbiota on bile acid metabolism is another factor influencing cholesterol absorption [40]. Gut microbiota influence the circulating LDL-C levels via the conjugation of primary bile acids to secondary bile acids [41] and by facilitating bile acid excretion into feces [40]. Specific bacterial phyla, such as *Lactobacillus* and *Clostridium*, can prevent the reabsorption of bile acids into the enterohepatic circulation through the deconjugation of bile acids. Since most bile acids are taken up in a conjugated form, deconjugation by gut microbes may be an important factor in mediating cholesterol levels. It is possible that a high-fat, low-carbohydrate diet alters gut microbiota composition in an unfavorable way so that the intestinal reabsorption of bile acids and cholesterol is enhanced, and hence, cholesterol accumulates more quickly in the circulation. Unfortunately, fecal samples were unavailable from the cases we described.

### 3.2. Increased Intake of Dietary Fatty Acids

In the second half of the 20th century, large epidemiological studies linked the composition of dietary fat with plasma LDL-C levels and cardiovascular outcomes [42]. It is now clearly established that the intake of saturated fatty acids (SFAs) increases the circulating LDL-C levels, although the overall effect of limiting SFA intake on cardiovascular health remains controversial [43,44]. Meta-regression analysis of 84 feeding studies showed that replacing the daily energy intake from SFAs with carbohydrates or unsaturated fatty acids considerably lowers LDL-C levels [45]. The LDL-increasing mechanism of SFAs was investigated in various in vitro and in vivo models [46] and likely involves decreased mRNA and protein expression of LDLR, as well as decreased LDLR activity [47]. It has to be noted that this effect depends on SFA type: 12–16 carbon fatty acids, found in dairy and red meat, have the largest effect to raise LDL-C [43,44]. This notion is supported by feeding trials in which increased SFA intake in the form of red meat resulted in higher LDL-C levels than the same SFA intake through nonmeat protein sources [48]. High SFA intake by our patients likely contributed to their hypercholesterolemia, and the fact that LDL-C levels dropped when patient 4 exchanged his intake of butter, meat, and dairy products for more vegetables and vegetable oils supports this notion.

Furthermore, an increased flux of fatty acids to the liver stimulates the assembly and secretion of VLDL particles [49], which would be in line with the abundance of VLDL-C observed with FPLC analysis in patients 1 and 2. However, the absence of triglycerides suggests other factors are also at play, such as extremely efficient lipolytic activity or the active exchange of cholesteryl esters.

### 3.3. Decreased Intake of Carbohydrates

Diets are generally considered very low in carbohydrates when the energy intake from carbohydrates is <10% of the total energy intake or <50 g per day [50]. These diets have been recommended as a treatment for patients with specific medical conditions, such as rare metabolic diseases or epilepsy, where hyperlipidemia is a well-known side effect of the ketogenic diet [51,52]. The ketogenic diet is gaining increasing scientific attention in the field of sports medicine, where multiple feeding trials have been conducted in healthy athletes that consistently show increases in LDL-C levels [53,54,55,56,57]. However, individual responses are variable, and LDL-C levels do not reach the levels consistent with homozygous FH, as observed in the extreme cases we presented. Carbohydrate-restricted diets have further gained attention through popular scientific news outlets and social media channels to reduce weight, as was the case for patient 4.

A decreased intake of carbohydrates inadvertently means that the relative intake of protein and/or fat is increased, but the relative importance of ‘low-carbohydrate’ vs. ‘high-fat’ diet intake in causing hypercholesterolemia remains to be established. It has been hypothesized that carbohydrate restriction leads to increased dependence on fat as a metabolic substrate, which drives the increased hepatic secretion of triglyceride rich VLDL. It is thought that triglycerides are taken up very rapidly by peripheral tissues, which have come to rely heavily on fats as their metabolic substrate. The resulting lipid profile is characterized by markedly elevated levels of LDL-C and HDL-C, yet low triglycerides [58]. Our observation of elevated VLDL-C, LDL-C, and HDL-C, combined with relatively low levels of triglycerides found in patients 1 and 2, are in support of this ‘lipid energy model’. The authors further hypothesize that low levels of insulin, as we observed in patients 1 and 2, in combination with low hepatic glycogen stores (not measured by us), contribute to increased VLDL secretion rates. There is anecdotal evidence that the reintroduction of carbohydrates reverses the hypercholesterolemia seen in these subjects [22,23], but this could not be accurately assessed in our patients. Future studies, such as stable isotope studies under controlled feeding conditions, are required to unravel the relative importance of carbohydrate restriction on the observed increases in LDL-C levels. Such studies can also be used to determine both the excretion and absorption rates of cholesterol in the gut, as well as the production and uptake of lipoproteins by the liver in the context of high-fat, low-carbohydrate diets.

## 4. Conclusions and Future Research

In conclusion, mounting evidence describes the relationship between high-fat, low-carbohydrate diets and severely elevated plasma LDL-C levels generally considered to be consistent with (homozygous) FH. This phenomenon provides unique opportunities to study the fundamental (patho)physiological mechanisms involving cholesterol and lipoprotein homeostasis. Clinicians should rule out high-fat, low-carbohydrate diets as a possible cause for hypercholesterolemia in patients presenting with clinical FH in whom no mutation is found and discuss dietary modifications to durably reduce LDL-C levels and ASCVD risk.

## Figures and Tables

**Figure 1 nutrients-15-00962-f001:**
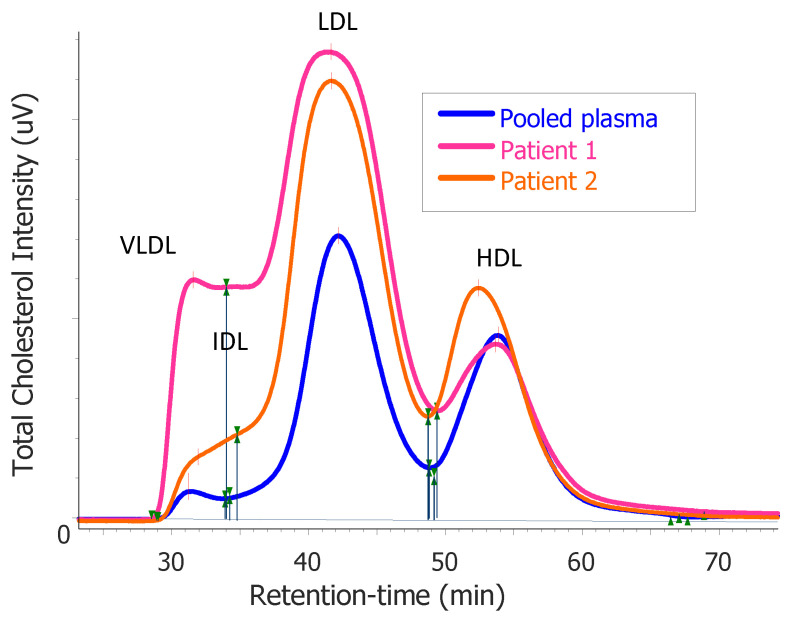
Results of the total cholesterol distribution in plasma after fast protein liquid chromatography (FPLC) profiling from patient 1 and patient 2 compared with that of pooled plasma from healthy subjects. The elution of the main lipoprotein classes is indicated for very-low-density lipoprotein (VLDL), intermediate-density lipoprotein (IDL), low-density lipoprotein (LDL), and high-density lipoprotein (HDL), respectively.

**Table 2 nutrients-15-00962-t002:** Lipid panels of patient 3.

	Referral	Lipid Clinic, First Visit	Lipid Clinic, Second Visit	Study Visit
Timepoint	T = 0	+3 Months	+6 Months	+4 Years
Total cholesterol (mmol/L)	15.0	10.07	5.11	3.79
HDL cholesterol (mmol/L)	2.22	1.76	1.55	1.17
LDL cholesterol (mmol/L)	12.15	7.93	3.40	2.29
Triglycerides (mmol/L)	1.40	0.85	0.35	0.74
ApoB (mg/dL)	N.A.	N.A.	N.A.	64

**Table 3 nutrients-15-00962-t003:** Lipid panels of patient 4.

	Pre-Diet	On Diet	Modified Diet
Timepoint	T = 0	+6 Months	+9 Months
Total cholesterol (mmol/L)	4.7	10.8	5.1
HDL cholesterol (mmol/L)	1.4	2.15	1.4
LDL cholesterol (mmol/L)	2.9	8.39	2.8
Triglycerides (mmol/L)	0.9	1.78	0.8
ApoB (mg/dL)	79	194	79

## Data Availability

Not applicable.

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
