# Peer review of "Severe Dyslipidemia Mimicking Familial Hypercholesterolemia Induced by High-Fat, Low-Carbohydrate Diets: A Critical Review"

_nutrients, 2023, doi:10.3390/nu15040962_

Round 1

Reviewer 1 Report

The manuscript is well-written. But, the manuscript looks like case report. The authors should organize the manuscript a bit more. My comments are as follows,

 Comments:

1. Line 21: Please describe several observations in Table, or, to clarify the subject and your suggestion, describing past (conventional ) attitudes, your opinion and how to solve the problems in Figure is better than Table.

2. Gut bacteria also affect bile acid metabolism and cholesterol uptake associated with hypercholesterolemia (Int J Mol Sci. 2021 Jul 28;22(15):8074.). For future research, please describe the interaction between bacteria and hypercholesterolemia in the text, and discuss. The bacterial interaction with hypercholesterolemia makes this review more attractive for readers.

3. line 315: please set “Increased” in small letter.

Author Response

Dear sir/madam,

We thank you for the response to our manuscript and the useful and constructive suggestions. Below, we provide a detailed response to all comments raised and point out the edits we made in response to them in our revised manuscript. Please also find our revised manuscript with tracked changes for your reference.

  1. Thank you for your relevant comment and suggestion to add information on gut microbiota and bile acid metabolism. We have now added a section discussing this topic in lines 299-309.
  2. Thank you for pointing out this topic for additional interesting discussion and for providing suggestions regarding relevant literature. We have now included a discussion on the role of gut bacteria in bile acid metabolism and possible implications in low-carbohydrate, high fat diet. We have supported this section with pertinent literature. Please see also lines 299 through 309.
  3. Thank you for the careful and attentive look at spelling, we have now corrected this mistake.

Reviewer 2 Report

The conclusion in this manuscript might be correct and I totally understand the big challenges to get enough numbers especially from human patients to darw the conclusion. However, with so small number of samples/patients, it is not sound or risky to get your conclusion.

Author Response

Dear sir/madam,

Thank you for the time and effort taken to review our work. Indeed, very little is known on this topic as of yet, and a limitation is indeed the small sample size. However, taking our observations together with a critical discussion of the literature, we do believe that our conclusions can be justified.

Our conclusions can be distilled as follows, as we have written as the final sentence of our abstract:

“Altogether, clinicians should rule out high-fat, low-carbohydrate diets as a possible cause for hypercholesterolemia in patients presenting with clinical FH in whom no mutation is found, and discuss dietary modifications to durably reduce LDL-C levels and cardiovascular disease risk”

Ruling out the adherence to high-fat, low-carbohydrate diet can be easily done when taking the patient’s history, without need for costly or time-consuming diagnostic approaches. Given the unequivocal, large body of evidence supporting the role of elevated plasma LDL-C in the development of cardiovascular disease, we believe that discussion of reducing one’s LDL-C is justified. Regarding potential (patho)physiological mechanisms linking hypercholesterolemia and high-fat, low-carbohydrate diet, we restrict ourselves to discussing potential mechanisms only with the backing of relevant literature from the field.

All in all, we hope this reviewer will follow us in the argument that limited sample size does not invalidate the conclusions drawn in this manuscript.

Reviewer 3 Report

Comments and Suggestions for Authors:

The article is generally very good, and interesting and provides knowledge that severe dyslipidemia replicates the effects of familial hypercholesterolemia induced by high-fat, low-carbohydrate diets. I recommend the publication of this study; however, it needs to be polished before acceptance. I would ask the authors:

1.      There are two types of lipoprotein that carry cholesterol. The low-density lipoprotein or LDL and high-density lipoprotein, or HDL. In the present study, the authors mentioned that reduced clearance of LDL particles from circulation is the cause of hypercholesterolemia. Do the authors perform any literature study or investigate whether the patient having impaired clearance of HDL also leads to hypercholesterolemia?

2.      If authors added some detail In Lines # 63-65, like “ while intake of processed meat, deep-fried foods, ice cream, yogurt, butter Coconut oil, cheese,  palm kernel oil, refined carbohydrates, and sweetened beverages high in saturated fatty acids (SFAs), cholesterol, trans fatty acids (TFAs), sodium and glucose should be avoided or kept to a minimum” will make more sense.

3.      How about the patient’s triglyceride level? Did you people perform tests for triglyceride levels as well?

4.      Which kind of dietary pattern you people have advised to the patients (Line # 109)? Please explain as this will provide extra knowledge to the scientific community.

5.      Liver makes about 80% of the cholesterol. Bad cholesterol is produced when there is any pathological condition/situation going on in the liver. Do you people perform any tests related to liver damage e.g. alanine aminotransferase (ALT) from these patients? Of course, you people did Vibration-controlled transient elastography (VCTE, also known as 'transient elastography'), but sometimes you people will need blood tests as well for liver damage because most of the time, particularly in  Abdominal ultrasonography liver looks normal with no focal lesions even in those persons who have a high ALT level.

6.      The conclusion is well but needs a broad explanation, it would be better if you people could summarize it more.

Author Response

Dear sir/madam,

Thank you for your time taken to review our work in detail. We have provided replies to your questions point by point below. Please also find the improvements made in the manuscript in a separate, revised file.

  1. Thank you for addressing the point of HDL. The reviewer is correct in saying that HDL is an important component of one’s total cholesterol. When comparing the cholesterol levels observed in our patients with the levels generally considered consistent with Familial Hypercholesterolemia, we have limited our discussion to levels of LDL-C only. This is because LDL-C is the most relevant in determining one’s cardiovascular disease risk. In the patients we describe, HDL-C levels were relatively high (1.53, 2.00, 2.22 and 1.40 mmol/L) and contributed to high levels of total cholesterol. However, the LDL-C fraction was of greater importance (levels >10mmol/L). We discuss high HDL-C in lines 353-356 as follows, and refer the the interested reader to relevant literature:

“The resulting lipid profile is characterized by markedly elevated levels of LDL-C and HDL-C, yet low triglycerides [56]. Our observation of elevated VLDL-C, LDL-C and HDL-C, combined with relatively low levels of triglycerides found in patients 1 and 2 are in support of this ‘lipid energy model’.”

  1. We like to thank the reviewer for this suggestion and have incorporated it in the sentence line number 63-64.
  2. Thank you for addressing this relevant point. In two of our patients (1 and 2), we found relatively low levels of triglycerides (0.57 and 0.65 mmol/L) and have presented these findings in table 1 on page 3. Low triglyceride levels were less pronounced in the other patients in our study. Taken together, we feel that our observations are insufficient to support strong conclusions regarding high-fat, low-carbohydrate diet on triglyceride levels. Nevertheless, we do discuss the topic in our discussion section as follows, and interested readers may find more information on this topic through the reference provided:

“It has been hypothesized that carbohydrate restriction leads to increased dependence on fat as a metabolic substrate, which drives increased hepatic secretion of triglyceride-rich VLDL. It is thought that triglycerides are taken up very rapidly by peripheral tissues, which have come to rely heavily on fats as their metabolic substrate. The resulting lipid profile is characterized by markedly elevated levels of LDL-C and HDL-C, yet low triglycerides [56].”

  1. Thank you for your pointing out this important omission. We have now clarified in lines 109 and 110 that we advised the patients first and foremost, to adopt a more balanced dietary pattern also containing carbohydrates (any source) and vegetables (any kind). We did not provide recommendations on the level of individual food products.
  2. We kindly thank the reviewer for raising a relevant point. We have tested liver enzyme (ALT, AST gGT and AF) levels and these were all found to be within the normal range. We have now clarified this in-text in lines 98 and 99.
  3. In order to sharpen our conclusion, we have now reduced our conclusion section by approximately 25% (please see section 4) and have moved relevant information elsewhere in the manuscript.

All in all, we hope to have answered the points raised by this reader satisfactorily and look forward to any further comments.

Round 2

Reviewer 2 Report

For pratical reasons, if the sample size can not be increased, the current version is acceptable. From scientific point of view, if the authors can get more data from other cohorts, definitely we need that.